

# Graph-based event schema induction in open-domain corpus

Keyu Yan[1,2], Wei Liu[1,2], Shaorong Xie[1] and Yan Peng[2,3,4]

[1] School of Computer Engineering and Science, Shanghai University, Shanghai, China
[2] Shanghai Artificial Intelligence Laboratory, Shanghai, China
[3] School of Future Technology, Shanghai University, Shanghai, China
[4] Institute of Artificial Intelligence, Shanghai University, Shanghai, China

## ABSTRACT

An event schema provides a formal language for representing events and modeling knowledge about the world. Existing event schema induction methods often only applies text features to the cluster, restricting its cluster capabilities. This article presents a Graph-Based Event Schema Induction model to extract structural features from our constructed graph. Inspired by in-context learning, we propose a way to conceptualize clusters to generate event schemas. We evaluated the clustering experiment using the Adjusted Rand Index (ARI), normalized mutual information (NMI), accuracy (ACC), and BCubed-F1 metrics and generated event schemas based on overlap ratio and acceptable ratio. The experimental results show that our method has shown improvement in terms of clustering effectiveness, and the generated event schemas achieved highly acceptable ratio.

## INTRODUCTION

An event is an action or occurrence taking place at a particular time and place involving one or more participants, which is one of the basic units for human beings to understand and experience the world (*Jackendoff, 1992*), such as *arrest*, *bombing*, and *election*. To represent events and model the world event knowledge, an event schema provides a conceptual, structural and formal language. Typically, an event schema is represented as an event type and a set of slots, where slots denote the roles involved in the event. For example, "*Type: bombing, Slots: perpetrator, victim, target, instrument*" (*Chambers & Jurafsky, 2011*).

However, event schemas are constructed with the assistance of experts, and the annotation process is expensive and time-consuming, such as Message Understanding Conference (MUC) (*Chinchor, Hirschman & Lewis, 1993*), Automatic Content Extraction (ACE) (*Doddington et al., 2004*) and TAC-KBP (*Ji & Grishman, 2011*). Since events are open-ended, new event types emerge in different domains. Therefore, we require event schema induction, to automatically generate event schemas of high quality and broad coverage.

Due to the openness, diversity, and sparsity of events in the real world, generating event schemas poses significant challenges. Firstly, there are numerous events, and new events

Corresponding author
Wei Liu, liuw@shu.edu.cn

are constantly emerging. Secondly, events in different domains exhibit diversity; similar events may have various linguistic expressions, and the semantics of event arguments could denote different events. Finally, event representations are often incomplete and lack event arguments.

To achieve event schema induction, mainstream methods can be divided into two categories: representation learning and clustering-based methods (*Chambers & Jurafsky, 2011*; *Yuan et al., 2018*; *Sha et al., 2016*), and probabilistic graphical model-based methods (*Cheung, Poon & Vanderwende, 2013*; *Chambers, 2013*; *Nguyen et al., 2015*; *Liu, Huang & Zhang, 2019*). These methods often adopt the bag-of-words representation, which lacks accuracy.

Some studies (*Li et al., 2021*; *Jin, Li & Ji, 2022*) proposed the concept of "complex event schema", a narrative event schema containing temporal order and multi-hop theoretical relations. It allows tracking event relations based on time, location, and evidence within an article. However, these studies typically operate at the document level, focusing more on the argument relations between events, and are unsuitable for short texts at the sentence level.

For sentence-level event type discovery models, *Shen et al. (2021)* defined event in the form of <predicate, object head>. It acquires a unified vector representation using a latent generative model and performs clustering to obtain event clusters. Although it can discover event types from the corpus, it cannot automatically generate the event schemas.

To address the limitations of *Shen et al. (2021)*, we propose a Graph-based model for Event Schema Induction (GESI). We divide the task of event schema induction into two subtasks: clustering and conceptualization.

The clustering task is utilized to discover similar events. Improving the effectiveness of a clustering model depends heavily on mining more features. With the use of diverse knowledge sources (*Speer, Chin & Havasi, 2017*), we discover a rich graph-based event feature to be extracted. Our model extracts each sentence's predicate verbs and entity nouns as graph nodes and builds explicit connections between nodes. Moreover, utilizing knowledge sources can reveal more implicit connections. We assume that similar events have similar structural features. Inspired by *Jin, Li & Ji (2022)*, we use Graph Autoencoder (*Kipf & Welling, 2016*) to encode structural features to discover similar events. Thereby enhancing the effectiveness of event clustering models.

The conceptualization task is a generalization of similar events, generating concept words from a collection of instance words. With the increase in model size and the expansion of corpus (*Devlin et al., 2019*; *Radford et al., 2019*; *Brown et al., 2020*; *Chowdhery et al., 2022*), large language models have demonstrated a new ability of in-context learning (*Min et al., 2022*). Many studies (*Wei et al., 2022b*; *Wei et al., 2022a*) show that in-context learning has become a new paradigm of natural language processing and can perform complex tasks by displaying a few examples composed of context. We discover that using in-context learning can help achieve event conceptualization tasks. Thus, we designed a prompt for event conceptualization that includes several demonstration examples, conceptualizing event clusters and generating event schemas, each consisting of an event type and multiple slots.

**Corpus**

During a large-scale protest in a major city, the police had to use their guns to safely arrest the protester.

During a violent riot in the downtown area, the cop used his spontoon to subdue the rebel who was inciting violence.

**Clustering**

{arrest, city, police, spontoon, rebel}
{subdue, downtown, cop, guns, protester}

**Conceptualization**

**Event Schema**

Type: Arrest
Slots: arrester: police/cop, arrestee: protester/subdue,
reason: protest/riot, location: city/downtown

**Figure 1** Clustering: discovery of similar events; conceptualization: conceptualizing similar events to generate event schemas, such as "Type: Arrest, slots: arrester, arrestee, reason, location", and append entity words of each slot from the corpus.

As shown in Fig. 1. The corpus is passed through a clustering task to obtain multiple clusters, where each cluster element is a set of words, with red denoting predicate verbs and blue denoting entity nouns. Then, the clusters go through a conceptualization task to generate an event schema consisting of an event type and multiple slots, with each slot also follows multiple entity nouns.

We conducted event mention clustering experiments, and compared with other models based on the four clustering metrics. The results show that our model is the highest in all metrics on the ACE 2005 and MAVEN-ERE datasets. We also conducted ablation experiments to prove the efficacy of different components within our model. Finally we evaluated the quality of the generated event schema by overlap ratio and acceptable ratio. although the generated event schemas exhibit less-than-ideal overlap ratio, they are still highly acceptable ratio.

The main contributions of this study are:

- We propose a graph construction method and extract event structure features from the graph to improve the clustering effect of the model.
- We are not limited to clustering for the event schema induction task and the conceptualization of similar events is well done with the help of in-context learning to generate event schema directly.
- We experimentally validate the effectiveness of the method. The structural features enhance the clustering model. And the generated event schemas have high Acceptable Ratio.

## RELATED WORKS

Early studies on event schema induction required knowledge of the templates and labeled datasets, including rule-based systems (*Chinchor, Hirschman & Lewis, 1993*; *Rau, Krupka & Jacobs, 1992*) and supervised classifiers (*Chieu, Ng & Lee, 2003*; *Bunescu & Mooney, 2004*; *Patwardhan & Riloff, 2009*). These classifiers utilized the surrounding context of the labeled examples for characteristics like nearby tokens, document position, syntax, named entities, semantic classes, and discourse relations. There are also ways (*Ji & Grishman, 2008*) to enhance labeled data with unlabeled data, by utilizing information retrieval to acquire a broader range of background knowledge from unlabeled knowledge sources.

Subsequently, people started using weakly supervised approaches to reduce the need for annotated data, typically by clustering documents specific to an event and extracting common word patterns as extractors (*Riloff & Schmelzenbach, 1998*; *Sudo, Sekine & Grishman, 2003*; *Riloff, Wiebe & Phillips, 2005*; *Patwardhan & Riloff, 2007*). *Filatova, Hatzivassiloglou & McKeown (2006)* assimilated named entities into pattern learning to approximate unspecified semantic roles. However, most methods still require the manual definition of templates and their slots.

In contrast, to remove these data assumptions, learning instead from a corpus of unknown events and unclustered documents without seed examples. People began to explore clustering methods based on representation learning and methods based on probabilistic graphical models.

In clustering methods for representation learning, *Chambers & Jurafsky (2011)* induced event schema as sets of linked events associated with semantic roles. They used pointwise mutual information and agglomerative clustering algorithms to cluster and extract role fillers from specific documents. *Yuan et al. (2018)* represented argument roles in event schema as <predicate:role:label> and learned embedding representations of argument role schema using entity co-occurrence information for clustering. They manually selected a subset of argument roles to construct event schema. *Sha et al. (2016)* proposed a joint entity-driven model to learn templates and slots simultaneously. They used the normalized cut criteria in image segmentation to divide the entities into template and slot clusters. *Shen et al. (2021)* extracted corpus data in the form of <predicate, object head> pairs, filtered and disambiguated it, generated respective vector representations using pre-trained BERT (*Devlin et al., 2019*) models, and learned a unified vector representation in a latent spherical space for clustering event types. Representation learning clustering methods mainly use

entity co-occurrence information to learn vector representations of argument roles and clusters to obtain event schemas. The disadvantage of this method is that it requires a reasonable definition of thematic role expressions or a large amount of high-quality data.

In probabilistic generative models, *Cheung, Poon & Vanderwende (2013)* used probabilistic methods to learn templates and event state transitions from text. *Chambers (2013)* induced schema by combining generative models with entity coreference. *Liu, Huang & Zhang (2019)* employed a neural latent variable model based on variational autoencoders to model event-type vectors and redundant information from entity mentions. They combined this with pre-trained ELMo language models (*Blei, Ng & Jordan, 2003*) to generate semantic features for entities and clustered entities in documents into several argument roles. Probabilistic generative methods mainly use generative or variational autoencoder models to learn event templates, argument roles, or event state transitions. The disadvantage of this method is that it uses entity clusters to represent argument roles, which have poor intuitiveness and interpretability.

Furthermore, *Li et al. (2021)* introduced the concept of "complex event schema", a comprehensive graph schema that includes temporal order and multi-hop causal relations. It allows tracking the relationships between events in an article based on time, location, and arguments. They employed an autoregressive graph generation model to model the first-order dependencies of event nodes concerning their neighbors. *Jin, Li & Ji (2022)* proposed a method for encoding global graph context using dual Graph Autoencoder to generate event skeletons. These methods primarily focus on modeling the relationships between multiple events within an article. The complex event schema falls under the category of narrative event schema. We mainly study event schema induction within the context of atomic event schema, with a concentration on discovering identical events and inducing event schemas.

In addition, there have been studies (*Wu et al., 2019*; *Zhao et al., 2021*; *Huang & Ji, 2020*) on open relation type extraction tasks that utilize known types to facilitate the learning of a similarity measure Subsequently, known type data is incorporated for joint training in the clustering process of unknown types. However, this method relies on well-annotated known types and only applies to discovering unknown types within a specific domain.

Our study is based on representation learning clustering methods. Unlike previous works, we divide the task into two subtasks: clustering and conceptualization. For the clustering task, our goal is to improve the clustering effect, thus extracting more event features. We attempt to discover structural information from the graph we constructed. Inspired by *Jin, Li & Ji (2022)*, we use Graph Autoencoder to encode this feature. For the conceptualization task, we drew inspiration from in-context learning. Accordingly, we propose a method to conceptualize clusters for the generation of event schemas.

## PROBLEM DEFINITION

Formally, given an unlabeled corpus, which has a set of event sentences $S = \{S_1, S_2, \ldots, S_n\}$, where each sentence $S_i = \{w_1, w_2, \ldots, w_m\}$ contains $m$ words and all words $w_i \in \mathbf{W}$, each sentence typically includes one predicate verbs $w_v \in \mathbf{W}$ and one or more entity nouns

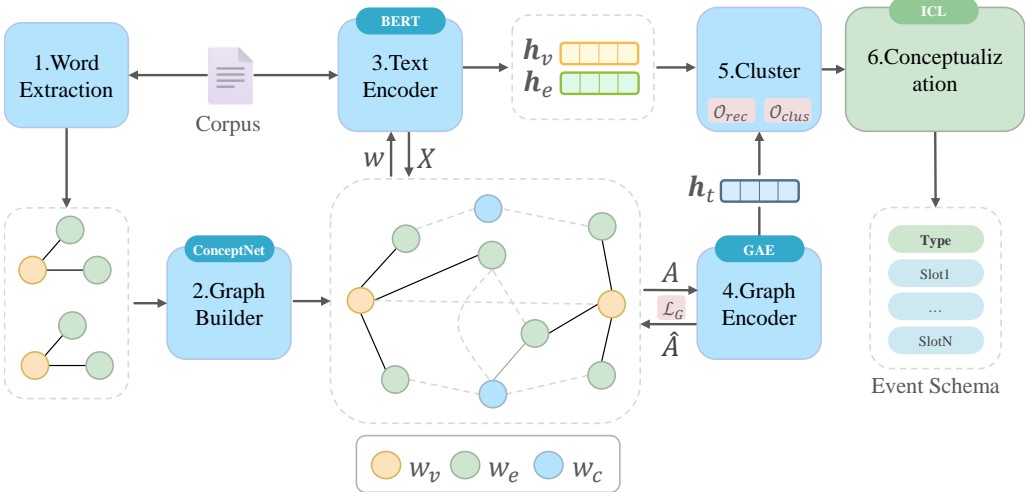

**Figure 2** **Overview of our event schema induction framework.** The input is a sentence-level corpus. To cluster, we use BERT and Graph Autoencoder as two feature encoders. Final conceptualization of clusters with the help of ICL (In-context learning). Event Schema is our final result.

$w_e \in W$. $W$ represents the set of words in all sentences. We divide task into two subtasks: (1) **Clustering**: The goal is to discover $k$ clusters $x = \{x_1, x_2, \ldots, x_k\}$ from $n$ sentences. (2) **Conceptualization**: The goal is to conceptualize $k$ clusters as $k$ event schemas $y = \{y_1, y_2, \ldots, y_k\}$. Each event schema $y_i$ is a text template consisting of an event type and multiple slots. Each slot can correspond to multiple entity words.

## METHODOLOGY

Our framework is based on representation learning clustering methods, which takes a sentence-level corpus as input and aims to output event schemas. The framework, as shown in Fig. 2, mainly consists of the following components: (1) Word Extraction to extract predicate verbs and entity nouns from the unlabeled corpus using dependency parsers, named entity recognition, filtering, and disambiguation techniques. Each sentence forms a word set $\{w_v, w_{e_1}, w_{e_2}, \ldots\}$, where $w_v$ represents the predicate verb and $w_{e_i}$ represents the entity noun. These word sets can be represented as subgraphs; (2) Graph Builder to connect all the subgraphs using the external knowledge base ConceptNet, forming a graph $G$; (3) Text Encoder to encode the semantic and contextual information of each word and generates $\mathbf{h}_v$ and $\mathbf{h}_e$. Also as the feature matrix $X$ for the graph encoder; (4) Graph Encoder to encode the structural feature of each subgraph in the graph. It utilizes a Graph Autoencoder and unsupervised training methods to iteratively reconstruct the adjacency matrix $A$ to learn the graph structure; (5) Clustering to unify all vector features from both the text encoder and graph encoder into a vector space and performs clustering; (6) Conceptualization to conceptualize each cluster as corresponding event schema by ICL (in-context learning) of large language model.
[1]We use OntoNotes sense grouping as our sense dictionary: https://verbs.colorado.edu/html_groupings/

## Word extraction

Word extraction is divided into three steps: (1) utilizing dependency parser and named reality body recognition to obtain candidate word sets; (2) filtering word sets based on salience formula; (3) word sense disambiguation (WSD).

### Extract candidate word sets

We adopted a corpus preprocessing method proposed by *Shen et al. (2021)*, which utilizes a dependency parser to analyze sentences $S_i$ and obtain the dependency parse tree, identifying the predicate verb $w_v$ and its target noun $w_{e_1}$ as the sentence's subject. Applying this method to all sentences in the corpus, we extracted a collection of $< w_v, w_{e_1} >$ pairs along with their frequencies. Additionally, we incorporated named entity recognition into this approach to extract other entity nouns $\{w_{e_2}, w_{e_3}, \ldots, w_{e_l}\}$ from the sentences, resulting in a set of entity nouns $\mathbf{w_e} = \{w_{e_1}, w_{e_2}, \ldots, w_{e_l}\}$ that are associated with the predicate $w_v$. Finally, in each sentence, we extract a candidate word set $\{w_v, w_{e_1}, w_{e_2}, \ldots\}$.

### Salience-based filter

The quality of the extracted predicate verbs and entity nouns varies significantly. Some are overly general and lack informative value, while others are infrequent and lack inductive value. We employed a formula (*Shen et al., 2021*) to measure the significance of words and filter their quality:

$$Salience(w) = (1 + \log(freq(w))^2) \log \frac{N_{bs}}{bsf(w)}$$

where $freq(w)$ is the frequency of word $w$, $N_{bs}$ is the number of background sentences, and $bsf(w)$ is the background sentence frequency of word $w$. This formula is based on the TF-IDF concept, considering two factors: the word should appear frequently in our corpus and not appear too frequently in large-scale general-domain background corpora. We selected the top 80% most significant words as the result.

### Word sense disambiguation

To address the issue of polysemy in predicate verbs, we introduced a sense dictionary[1] and combined it with the word sense disambiguation model LMGC (*Wahle et al., 2021*). We input the instances of the predicate in the corpus, denoted as $S_i$, along with example sentences $S_i'$ for each verb sense into the model. The model outputs the most similar example sentence corresponding to the current predicate sense.

## Graph builder

To enhance the effectiveness of clustering models, more features are usually required. We find a rich graph-based event feature that can be extracted using external knowledge sources (*Fan et al., 2022*). In this section we describe how we construct the features of the graph.

We transform each $\{w_v, w_{e_1}, w_{e_2}, \ldots\}$ set into a subgraph $\mathbf{G_i}$. Each word corresponds to a node. Edges $(w_v, w_e)$ connecting predicate verbs $w_v$ with entity nouns $w_e$, indicating the association between the predicate verb and the entity noun.

We process all subgraphs $\mathbf{G_i}$ as input and generate a graph $\mathbf{G}$ using Algorithm 1. The first step of the algorithm is to concatenate all subgraphs. We iterate over each node in the subgraph set $\{\mathbf{G_1}, \mathbf{G_2}, \ldots, \mathbf{G_n}\}$ and store the nodes as keys in the dictionary. The corresponding values are lists that store the sentence IDs where the key node appears. Then, we use a list to store all the edges.

---

**Algorithm 1** : Graph Builder.

---

**Input:** Subgraph list $[\mathbf{G_1}, \mathbf{G_2}, \ldots, \mathbf{G_n}]$, $\mathbf{G_i} = [\mathbf{nodes_i}, \mathbf{links_i}]$;
    where $\mathbf{nodes_i}$ is a list, contains predicate verb and entity nouns of sentence $\mathbf{S_i}$, $\mathbf{links_i}$ is a lists, each element consists of
    two node ids, indicating that there is a correlation between the two;

1:  $\mathbf{G} \leftarrow \varnothing$;
2:  $\mathbf{nodes} \leftarrow \varnothing$;  // A hashmap, key is str, value is list.
3:  $\mathbf{links} \leftarrow \varnothing$;  // A list.
4:  // Step1. Concat all subgraphs
5:  **for** i $\leftarrow$ 0 **to** $n \in$ **do**
6:    **for** j $\leftarrow$ 0 **to** $Len(G_i[0])$ **do**
7:      $node = G_i[0]_j$;
8:      **if** node not in **nodes then**
9:        $\mathbf{nodes} \leftarrow Put(node, [i])$;
10:     **else**
11:       $\mathbf{nodes} \leftarrow Append(\mathbf{links}, G_i[1])$;
12:     **end if**
13:    **end for**
14:   $\mathbf{links} \leftarrow Append(\mathbf{links}, \mathbf{G_i}[1])$
15:  **end for**
16:  // Step2. Query and link concept node by ConceptNet WebAPI
17:  **for** i $\leftarrow$ 0 to $Len(\mathbf{nodes})$ **do**
18:    $node, sentIds = Entry(\mathbf{nodes_i})$;
19:    $\mathbf{edges} = Query(node)$;   // request WebAPI.
20:    **for** j $\leftarrow$ 0 **to** $Len(\mathbf{edges})$ **do**
21:     $conceptNode = edges_i[end]$;
22:     **if** $conceptNode$ **not in nodes then**
23:       $\mathbf{nodes} \leftarrow PutIfAbsent(conceptNode)$;
24:       $\mathbf{links} \leftarrow Append([node, conceptNode])$;
25:     **end if**
26:    **end for**
27:  **end for**
28:  $\mathbf{G} = [\mathbf{nodes}, \mathbf{links}]$
**Return:**  $\mathbf{G}$;

---

The second step of the algorithm is to iterate the previous nodes. Each node serves as a query keyword to request responses from the ConceptNet WebAPI. The responses from the API provide the edges associated with the current node in ConceptNet. These edges contain information about related words and their relationships with the current node. The official documentation of ConceptNet WebAPI (https://github.com/commonsense/conceptnet5/wiki/Relations) includes 34 types of relationships. We exclude unreasonable associations, such as "*ExternalURL*" and "*Antonym*". We iterate over these edges, extract the associated words (concept words), add them to the set, and append their corresponding edges to the list.

Finally, we merge all nodes and links to form the result $\mathbf{G}$.

## Text encoder

To extract features for each word $w$ (predicate verb $w_v$ or entity noun $w_e$), the most basic method is to encode the text. Two distinct features can be computed: contextual

and semantic features: (1) Contextual feature refers to the information surrounding a particular word or sequence of words within a text. (2) Semantic feature denotes the meaning attributes associated with a specific word. We computed semantic features to represent the meaning, as words may have different senses. Finally, these contextual and semantic features are aggregated to form the textual feature representation for the word.

For the textual feature. Firstly, we locate the word $w$ in the corpus. During the graph building, each node records the sentences in which the current word appears. Next, we identify the position of the word $w$ in the token sequence of each sentence. We extract the $M$ tokens before and after the word $w$ as the context. We input the context token sequence into BERT and take the CLS token as the contextual feature $\mathbf{h}_{w,i}^{ctx}$ for word $w$ in the $i$th sentence.

For the semantic feature. Firstly, vectorize the interpretation of each word in the sense dictionary. and then input it into BERT, taking CLS as the semantic features $\mathbf{h}_w^{ses}$.

Finally, we average all the contextual features of word $w$ to obtain $\mathbf{h}_w^{txt}$:

$$\mathbf{h}_w^{txt} = [\mathbf{h}_{w,i}^{ses}; \mathbf{h}_w^{ctx}].$$

## Graph autoencoder

This section will explain how to use an unsupervised graph neural network model to encode structural features for each node in the graph **G**. We use the Graph Autoencoder model for unsupervised graph representation learning. Figure 3 shows that graph autoencoder consists of an encoder and a decoder. The encoder learns hidden representations $\mathbf{h}_t$ of nodes from the raw graph data while the decoder reconstructs the original graph structure from the hidden representations $\mathbf{h}_t$:

$$\mathbf{h}_t = \widetilde{A}\,\mathbf{RELU}(\widetilde{A}XW^0)W^1$$

where the adjacency matrix $A$ and the feature matrix $X$ are the inputs to the model. $\mathbf{h}_t \in \mathbb{R}^{N \times d}$, $N$ is the number of nodes. $d$ is the dimension. $\widetilde{A} = D^{-\frac{1}{2}}AD^{-\frac{1}{2}}$ represents the symmetrically normalized adjacency matrix. And $W^0$ and $W^1$ are the GCN parameters that need to be learned. For the adjacency matrix $A$, we can obtain it directly from the graph **G**.

For the feature matrix $X$, we use $\mathbf{H}_w^{txt}$ obtained from the text encoder as the feature $X_i$.

For the concept $w_c$ without $\mathbf{H}_w^{txt}$, we use a context replacement method. We randomly sample a neighboring node of the concept word $w_p$. Then, we replace the instance word $w_p$ in sentence $S_p = \{w_1, w_2, \ldots, w_p, \ldots, w_n\}$ with the concept word $w_c$ to generate a pseudo-sentence $S_c' = \{w_1, w_2, \ldots, w_c, \ldots, w_n\}$. This pseudo-sentence is regarded as the context for the concept word $w_p$. Finally, the pseudo-sentence and the concept word are fed into the text encoder, and the resulting output is used as the feature $X_i$.

Then, the decoder reconstructs the adjacency matrix through the inner product of the embedding vectors $\hat{A}$:

$$\hat{A} = \sigma(\mathbf{h}_t \mathbf{h}_t^T).$$

In Graph Autoencoder, our goal is to optimize the parameters $W^0$ and $W^1$ of the encoder to reconstruct the adjacency matrix $\hat{A}$ using the decoder, aiming to make it as similar

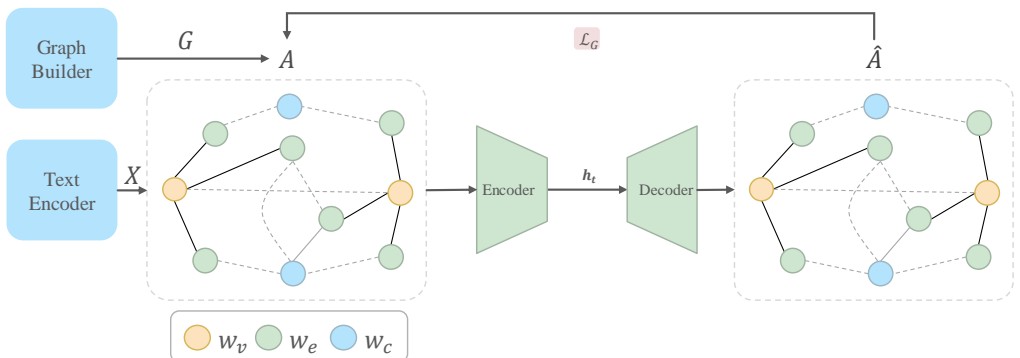

**Figure 3  The architecture of Graph Autoencoder module.**

as possible to the original adjacency matrix $A$. Since the adjacency matrix determines the graph's structure, the closer the reconstructed adjacency matrix, based on the node embedding vectors, is to the original adjacency matrix, the better the node embedding vectors represent the graph structure.

The loss function of Graph Autoencoder is defined as follows:

$$\mathcal{L}_G = -\frac{1}{N}\sum(y\log\hat{y}+(1-y)\log(1-\hat{y}))$$

where $y$ denotes an element in the original adjacency matrix $A$, taking values of 0 or 1. A value of 1 indicates the presence of an edge between nodes, whereas 0 indicates its absence. $\hat{y}$ denotes an element within the reconstructed adjacency matrix $\hat{A}$. By minimizing the cross-entropy loss between the reconstructed adjacency matrix and the original adjacency matrix, the Graph Autoencoder model can learn hidden node representations that preserve the structural information of the graph. We utilize this model to learn node representations in the context of the graph.

## Clustering

The goal of this section is to cluster the word set $\{w_v, w_{e_1}, w_{e_2}, \ldots\}$ as a clustering unit. The feature representations required for word sets are categorized into three types, namely verb textual features $\mathbf{h}_v$, entity noun textual feature $\mathbf{h}_e$ and structural features $\mathbf{h}_t$.

To cluster the three types of features, it is not feasible to simply aggregate them as $[\mathbf{h}_v; \mathbf{h}_e; \mathbf{h}_t]$. We need to unify all the features into the same vector space. To achieve this, we adopted a Latent Space Generative Model (*Shen et al., 2021*), which can unify two feature spaces and guide the clustering process based on the clustering objective. The clustering process can benefit from the well-separated structure in the latent space, resulting in mutual reinforcement. We modified this network model to handle the three types of features.

The model has two objective functions. The first objective is the reconstruct objective $\mathcal{O}_{\text{rec}}$ so that the model retains as much of the semantics of the input space as possible:

$$\mathcal{O}_{\text{rec}} = \sum_{i=1}^{N}\Big(\cos(\mathbf{h}_{v_i}, g_v(f_v(\mathbf{h}_{v_i}))) + \cos(\mathbf{h}_{e_i}, g_e(f_e(\mathbf{h}_{e_i}))) + \cos(\mathbf{h}_{t_i}, g_t(f_t(\mathbf{h}_{t_i})))\Big)$$

where $N$ denotes the number of cluster units. $g_v, g_e, g_t$ denotes the deep neural network. Joint learning mapping functions $f_v : \mathbf{h}_v \to \mathbf{Z}$, $f_e : \mathbf{h}_e \to \mathbf{Z}$ and $f_t : \mathbf{h}_t \to \mathbf{Z}.\mathbf{Z}$ is the vector space. The second objective is the clustering-promoting objective $\mathcal{O}_{\text{clus}}$. Assumes the existence of a spherical potential space with K clusters. Each cluster in this space corresponds to the same events and is associated with a von Mises-Fisher (vMF) distribution (*Banerjee et al., 2005*).

The vMF distribution of the event cluster is determined by the mean vector **c**, and the goal of the model is to learn the latent space with K well-separated cluster structures. Specifically, an expectation–maximization algorithm $p(\cdot)$ is used to sharpen the posterior distribution of each word set. The cluster soft assignment $q(\cdot)$ for each word set is computed and used to update the model during the maximization step. So the clustering objective function $\mathcal{O}_{\text{clus}}$ can be obtained as follows:

$$\mathcal{O}_{\text{clus}} = \sum_{i=1}^{N} \sum_{k=1}^{K} q(c_k|\mathbf{z}_i) \log p(c_k|\mathbf{z}_i)$$

where $N$ is the number of word sets, $K$ is the number of clusters. $c_k$ is sampled from the joint distribution and the implicit vector $z_i$ is generated from the vMF distribution. We first train the model during training using only the first objective function. Then, we jointly train the model using all the objective functions $\mathcal{O}$:

$$\mathcal{O} = \mathcal{O}_{\text{rec}} + \lambda \mathcal{O}_{\text{clus}}$$

where the hyperparameter $\lambda$ is used to balance the two objective functions. After clustering with this model, we obtain $k$ clusters. We keep each group's top $m$ cluster elements, and each cluster element is mapped back to the original set $\{w_v, w_{e_1}, w_{e_2}, \ldots\}$ for event schema induction.

## Conceptualization

To conceptualize clusters and generate event schemas, traditional methods manually cause possible slots from clustered event mentions or simply divide slots into *Person*, *Org*, *Physical Object*, or *Other* and then classify entity words. The disadvantage is the high cost associated with manual construction.

As the scale of models and corpus expands, large language models like GPT-3 are trained on amounts of internet text data. They predict and generate the next token based on a given context. The combination of large datasets and high-parameter language models has produced compelling language models and developed a new ability known as in-context learning (*Min et al., 2022*), which has become a new paradigm in natural language processing. Excellent results can be achieved by demonstrating examples composed of a few contexts.

So, to conceptualize each cluster into a corresponding event schema, we use in-context learning to model conceptualization as an in-context generation process.

Specifically, we design a prompt as shown in Table 1. The "Example" section includes five event sentences describing the Bombing event, the event type, and slots. For each slot, a set of words is appended to indicate the specific entities referred to by that slot in the sentences

**Table 1  Illustration of in-context learning.** Taking the demonstration example and a target as the input, large language models are responsible for making predictions.

| | Prompt |
|---|---|
| Example | Give you multiple sentences to generate an event schema. Examples: <br><br> Sentences: $\{S_1^{'}, S_2^{'}, ...\}$ <br> EventType: Bombing <br> Slots: perpetrator: terrorist group/individuals; <br> target: building/infrastructure/vehicle; <br> casualty: deaths/injuries; <br> location: country/city; <br> ... |
| Target | Now, generate a event schema by these sentences: $\{S_1^{'}, S_2^{'}..., S_5^{'}\}$ |

above. The sentences provided in the "Target" section are the sentences to be induced. We assume that each cluster contains only one event type, so in the "Target" section, we require it to generate one event schema at a time. In this process, the granularity of event schemas depends on the clustering algorithm and parameters used. Fine-grained settings may generate event schemas corresponding to specific event types, while coarse-grained settings may generate more general event schemas.

## EXPERIMENTS

We conduct experiments for the clustering task and the event schema induction (conceptualization) task, respectively, and we describe the datasets used, the evaluation metrics, the experimental setting, and the baseline models. For the clustering task, we compare all event clustering related baseline class models. Then, ablation experiments were conducted to demonstrate the effectiveness of each module. For the event schema induction task, we show and assess the event schemas generated.

In the following experiments, we refer to our model as GESI to stand for "Graph-Based Event Schema Induction in Open-Domain Corpus."

### Datasets and evaluation metric

*Datasets.*  We validated the effectiveness of the GESI model on the ACE 2005 (*Doddington et al., 2004*) and MAVEN-ERE (*Wang et al., 2022*) datasets. For each dataset, we followed the same preprocessing steps described in previous work (*Lin et al., 2020*; *Li, Ji & Han, 2021*). The ACE dataset consists of 17,172 sentences and covers 33 event types, while the MAVEN-ERE dataset consists of 29,748 sentences and covers 167 event types.

*Cluster Metrics.*  The following metrics for cluster quality evaluation are adopted: (1) NMI denotes the normalized mutual information between two cluster assignments. (2) ARI (*Hubert & Arabie, 1985*) measures the similarity between two cluster assignments based on the number of pairs in the same/different clusters. (3) ACC measures the clustering quality by finding the maximal matching between the predicted clusters and the ground

truth clusters using the Jonker-Volgenant algorithm (*Crouse, 2016*). (4) BCubed-F1 (*Bagga & Baldwin, 1998*) estimates the quality of the generated cluster assignment by aggregating the precision and recall of each element.

*Induction metrics.* Evaluating the goodness of event schemas under the labeled dataset is essential to determine the degree of matching between event types and event slots. We used two metrics to evaluate the event schemas. Two metrics that can be applied to event types and event slots, respectively. (1) Overlap ratio: The overlap ratio of event types refer to the proportion of correct event types to all generated event types. The overlap ratio of event slots refer to the mean value of the proportion of correct event slots among the generated event roles; (2) Acceptable ratio: The acceptable ratio of event types refer to the proportion of acceptable event types to all generated event types. The Acceptable Ratio of event slots refer to the mean value of the proportion of acceptable event slots among the generated event roles.

## Experimental setting

Our model comprises six main components: (1) Word Extraction: We use the en_core_web_lg model from spaCy 3.5 for dependency parsing and named entity recognition, and then utilize a verb sense dictionary based on OntoNotes sense grouping to filter irrelevant meanings and disambiguation. This dictionary is based on OntoNotes sense grouping. (2) Graph builder: We employ the WebAPI of ConceptNet, an external knowledge base, to establish word connections. (3) Text encoder: We adopt the bert-base-uncased model from BERT to encode the original text. (4) Graph encoder: We employ Graph Autoencoder, an unsupervised graph embedding learning model, with the input dimensions of the first GCN layer set to 512 and the second layer to 256. The learning rate is set to 0.001, and the training epoch is set to 250. (5) Clustering model: We utilize the Latent Space Generative Model for feature unification and clustering. The dimension parameters are set to 1000-2000-1000-100, and the aggregation strategy is based on the product method. (6) Conceptualization: We utilize the API provided by Claude to access its model, which is a large-scale PLM similar to GPT-3.

## Baselines

We validate the effectiveness of our proposed GESI using the following baselines. (1) Triframes (*Ustalov et al., 2018*): A graph-based clustering algorithm that constructs a k-NN event mention graph and uses a fuzzy graph clustering algorithm WATSET (*Ustalov et al., 2019*) to generate the clusters. (2) Joint constrained spectral clustering (JCSC) (*Huang et al., 2016*): A joint constrained spectral clustering method that iteratively refines the clustering result with a constraint function to enforce inter-dependent predicates and objects to have coherent clusters. (3) ETypeClus (*Shen et al., 2021*): A latent space joint embedding and clustering algorithm.

## Results of event mention clustering

Tables 2 and 3 present our model's comparative clustering evaluation results and the baseline model on the ACE 2005 and MAVEN-ERE event datasets, respectively. All values

**Table 2  Event mention clustering results on dataset ACE 2005.**

| Methods | ACE 2005 | | | |
|---|---|---|---|---|
| | ARI (std) | NMI (std) | ACC (std) | BCubed-F1 (std) |
| Triframes | 19.35 (6.60) | 36.38 (4.91) | — | 38.91 (2.36) |
| JCSC | 36.10 (4.96) | 49.50 (2.70) | 46.17 (3.64) | 43.83 (3.17) |
| ETypeClus | 40.78 (3.20) | 57.57 (2.40) | 48.35 (2.55) | 51.58 (2.50) |
| GESI | 54.08 (1.01) | 62.02 (0.75) | 50.01 (0.83) | 58.70 (1.52) |

**Table 3  Event mention clustering results on dataset MAVEN-ERE.**

| Methods | MAVEN-ERE | | | |
|---|---|---|---|---|
| | ARI (std) | NMI (std) | ACC (std) | BCubed-F1 (std) |
| Triframes | 22.47 (3.57) | 41.09 (3.44) | — | 44.59 (2.82) |
| JCSC | 40.54 (2.39) | 54.45 (1.57) | 49.16 (1.62) | 47.72 (1.33) |
| ETypeClus | 47.62 (1.69) | 60.79 (1.54) | 51.72 (0.76) | 52.09 (0.86) |
| GESI | 55.61 (1.05) | 67.51 (1.30) | 54.31 (1.78) | 57.49 (0.66) |

are expressed in percentages. We conducted each method ten times, presenting the final results as the average and standard deviation for each metric. Please note that the ACC metric does not apply to Triframes as it assumes an equal number of predicted and ground truth results in clusters.

The comparative results indicate that, on the ACE 2005 dataset, compared to the best-performing model, GESI achieved improvements of 13.3%, 4.45%, 1.66%, and 7.12% in ARI, NMI, ACC, and BCubed-F1, respectively. On the MAVEN-ERE dataset, compared to the best-performing model, GESI demonstrated improvements of 7.99%, 6.72%, 2.59%, and 5.4% in ARI, NMI, ACC, and BCubed-F1, respectively.

## Ablation study of event mention clustering

To further verify the effectiveness of filtering, disambiguation, and graph autoencoders in the event induction task, we conducted ablation experiments on the datasets ACE 2005 and MAVEN-ERE with the complete model and several variants to better understand their relative importance.

- w/o $\mathbf{h}_t$: Removing the graph autoencoder model means that graph structure embedding features are not generated, only text embedding features of predicates and entity nouns.
- w/o Filter: Removing the filter means that the significance of the extracted words will not be considered, and the quality of the words may vary.
- w/o WSD: Removing the word sense disambiguation model means that the polysemy problem of predicate verbs will not be considered, and polysemous words will be clustered as the same word.

During a series of experiments, removing a certain part and keeping the rest unchanged, the following observations can be drawn from Tables 4 and 5.

**Table 4  Ablation results on dataset ACE 2005.**

| Ablation | ACE 2005 | | | |
|---|---|---|---|---|
| | ARI (std) | NMI (std) | ACC (std) | BCubed-F1 (std) |
| GESI | 54.08 (1.01) | 62.02 (0.75) | 50.01 (0.83) | 58.70 (1.52) |
| w/o $h_t$ | 50.08 (0.99) | 56.99 (1.02) | 46.70 (1.30) | 52.96 (1.17) |
| w/o Filter | 52.40 (1.38) | 57.46 (1.37) | 48.24 (1.17) | 53.91 (1.25) |
| w/o WSD | 51.24 (0.85) | 57.14 (0.84) | 47.16 (0.75) | 55.35 (1.34) |

**Table 5  Ablation results on dataset MAVEN-ERE.**

| Ablation | MAVEN-ERE | | | |
|---|---|---|---|---|
| | ARI (std) | NMI (std) | ACC (std) | BCubed-F1 (std) |
| GESI | 55.61 (1.05) | 67.51 (1.30) | 54.31 (1.78) | 57.49 (0.66) |
| w/o $h_t$ | 44.73 (1.38) | 60.88 (1.32) | 46.79 (0.71) | 50.72 (0.85) |
| w/o Filter | 45.80 (0.75) | 61.91 (0.84) | 48.25 (0.79) | 51.75 (1.30) |
| w/o WSD | 46.79 (0.79) | 62.74 (0.29) | 49.62 (1.30) | 52.87 (1.27) |

- As can be seen from the absolute percentages of the different model evaluation metrics in Tables 4 and 5, the highest scores are for the complete GESI model. Removing any component of the model results in a decrease in performance, which confirms that the inclusion of graph autoencoders, filtering, and disambiguation can all improve the clustering ability of the model.
- From the relative changes of the different variants of the model, it can be seen that the largest performance degradation is in $w/o$ $h_t$. This indicates that the structural features generated by the graph autoencoder have a greater effect on model enhancement.
- From the comparison of each variant model with the complete GESI model, it can be seen that the w/o Filter variant model performs better than the w/o WSD variant model on the ACE 2005 dataset, and the opposite is true on the MAVEN-ERE dataset, suggesting that filtering is more important than disambiguation in the presence of large data sizes.
- In terms of the performance of the model on different datasets, for example, the variant model $w/o$ $h_t$, ARI, NMI, ACC, and BCubed-F1 decreased by about 4%, 5.03%, 3.31%, 5.74% on ACE 2005 dataset. And on MAVEN-ERE dataset, ARI, NMI, ACC, and BCubed-F1 decreased by about 10.88%, 6.63%, 7.52%, and 6.77% respectively, after removing the graph autoencoder, of which ARI decreased more significantly. Compared with the ACE 2005 dataset, ARI decreased more than other metrics, indicating that the graph autoencoder significantly improved ARI for larger datasets.

In summary, each component of the model in this article contributes to the final results, with slight variations in the performance of different components in different datasets, but overall a positive effect enhancement, especially for the structural information generated by the graph autocoder.

**Table 6  Sample of event schema induction.**

| Type: Arrest | Type: Conflict |
|---|---|
| arrester: police/security officials | parties involved: countries/groups |
| arrestee: individuals/suspects | weaponry: artillery/aircraft |
| reason: bombing/terrorism/drug charge | reason: territorial/resources |
| location: country/city | location: country |

**Table 7  Sample of event schema induction that are not explicitly labeled in the dataset.**

| Type: Aid | Type: Space Exploration |
|---|---|
| provider: countries/troops | agency: NASA |
| recipient: civilians | mission: mars rover |
| supplies: food/water | goal: search for water |
| location: country | launch date |

**Table 8  Schema induction performance result.**

| Method | Dataset | Event-type | | Event-argument | |
|---|---|---|---|---|---|
| | | Overlap ratio | Acceptable ratio | Overlap ratio | Acceptable ratio |
| ETypeClus | ACE 2005 | 17.16% | 78.45% | 6.64% | 47.86% |
| GESI | | 18.31% | 80.14% | 7.13% | 48.34% |
| ETypeClus | MAVEN-ERE | 53.95% | 79.56% | 34.41% | 76.70% |
| GESI | | 55.63% | 80.00% | 36.82% | 78.69% |

## Results of event schema induction

In Tables 6 and 7, we present sample of the results of our model's event schema induction. Our model successfully identifies the most true event schemas and produces highly accurate slots and detailed entity participants. It correctly identifies most real event schemas, such as Arrest and Conflict, and also identifies event types that are not explicitly labeled in the dataset, such as Aid and Space Exploration. This indicates that our model is highly effective and reliable. These results make our model well-suited for open-domain event schema induction research and its subsequent downstream applications.

Table 8 illustrates the performance of event schema induction. We observed that in the ACE 2005 dataset, GESI covered 18.31% of event types, with almost all uncovered event types being deemed acceptable (80.14%). However, for event slots, only 7.13% were covered, potentially due to the diversity and sparsity of event slots. In the MAVEN-ERE dataset, we observed that the model performs better, likely attributed to the larger scale of the MAVEN-ERE data, which reduced the sparsity of the data. We also conducted experiments on clusters generated by the ETypeClus method. The experimental results show that the method with better clustering performs better in event schema induction.

# CONCLUSIONS AND FUTURE WORK

This work explores a graph-based model for event schema induction. We divide the task of event schema induction into two subtasks: clustering and conceptualization. In cluster tasks, to enhance the clustering performance, we leverage external knowledge sources to discover structural information between words and employ a graph autoencoder for feature encoding. Experiments in event clustering show that structural features improve clustering outcomes across four clustering metrics. In conceptualization tasks, inspired by in-context learning, we used in-context learning to conceptualize similar events within clusters. We evaluated quantitatively using overlap ratio and acceptable ratio. Although the generated event schemas exhibit less-than-ideal overlap ratio, they still achieve highly acceptable ratio.

However, there are certain limitations in our research. In the future, we suggest exploring the following directions: (1) Using weighted or heterogeneous graphs in graph learning may yield better results because it can show more features. (2) The clustering model currently requires manual specification of the number of clusters. Exploring the automatic determination of the cluster number is a worthwhile direction to pursue. (3) The framework involves multiple sub-tasks. And joint training could reduce error propagation.

## Funding

This work was supported by the Major Program of the National Natural Science Foundation of China (No. 61991410) and the Program of the Pujiang National Laboratory (No. P22KN00391). The funders had no role in study design, data collection and analysis, decision to publish, or preparation of the manuscript.

## Grant Disclosures

The following grant information was disclosed by the authors:
The Major Program of the National Natural Science Foundation of China: No. 61991410.
The Program of the Pujiang National Laboratory: No. P22KN00391.

## Competing Interests

The authors declare there are no competing interests.

## Author Contributions

- Keyu Yan conceived and designed the experiments, performed the experiments, analyzed the data, performed the computation work, prepared figures and/or tables, and approved the final draft.
- Wei Liu conceived and designed the experiments, analyzed the data, authored or reviewed drafts of the article, and approved the final draft.
- Shaorong Xie analyzed the data, authored or reviewed drafts of the article, and approved the final draft.
- Yan Peng conceived and designed the experiments, authored or reviewed drafts of the article, and approved the final draft.

## Data Availability

The code is available in the Supplemental File.

The MAVEN-ERE datasets are available at GitHub: https://github.com/THU-KEG/MAVEN-ERE.

The ACE 2005 datasets are available at: https://catalog.ldc.upenn.edu/LDC2006T06.

## Supplemental Information

Supplemental information for this article can be found online at http://dx.doi.org/10.7717/peerj-cs.2155#supplemental-information.

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
