# Peer review of "Graph-based event schema induction in open-domain corpus"

_PeerJ Computer Science, doi:10.7717/peerj-cs.2155_

## Round 0.1 · original submission · Major Revisions

Please take a careful look at the feedback from the reviewers and consider addressing the issues raised, in particular, consider revising:

1. The introduction with clear definitions and stronger citations to back claims that are used to motivate the work.
2. The experimental section to be more focused and easier to follow. Do formulate the problem clearly and summarize the results in terms of how the original problem is validated.
3. Make an editorial pass over the text and presentation to correct errors, as outlined by the reviewers.

**Language Note:** The Academic Editor has identified that the English language must be improved. PeerJ can provide language editing services - please contact us at [email protected] for pricing (be sure to provide your manuscript number and title). Alternatively, you should make your own arrangements to improve the language quality and provide details in your response letter. – PeerJ Staff

·

Basic reporting

The methodology is well presented but it still has issues that need to be addressed. The introduction and conclusion need a considerable amount of editing to make it appropriate. What follows are the language and flow issues that I encountered:

Page 1: The first sentence defines an event using language that is easy to understand for non-experts. The rest of the paragraph introduces many domain concepts that must be defined. The paragraph could use some work to improve the flow and explanation of concepts such as class of events, argument role representation, etc.

Page 2: The paragraph that starts at line 50 needs to be shorter and smoother. There are several ideas intertwined and it is difficult to understand. The paragraph should be broken down into steps, for example, separate the discussion of the proposed model from the issues with event features. The discussion over edge excavation should be its paragraph and the same for the scaling issues.

The introduction, abstract, and conclusion use ambiguous language when describing the contribution. A summary of the experimental results that support the conclusion should be added. The paragraph at line 78 says the result shows this model is better than traditional clustering but doesn’t provide any reference point to what “better” means.

The sentences in lines 50 and 52 are very similar and redundant and can be condensed into a single sentence. The same for the conclusion where these sentences are repeated.

The sentence that starts at line 98 should dive a little deeper into the work of Ji and Grisham (one sentence that explains their work)

The sentence in line 101 is unclear. The references should be at the end of the sentence to improve readability but with the references removed, the sentence makes little sense.

Multi-word concepts should be broken down. For example, in line 11, instead of representation learning clustering methods, it would be best to write it as clustering methods for representation learning.

Line 133, the citation year has an extra letter.

Line 190, please provide the significance formula mentioned and briefly explain it.

Lines 199 to 213. There are two steps for feature representation and the first step has two steps on its own. These two paragraphs are very confusing to read. Maybe present an overview of steps and substeps using a list with bullet points. The text states that step two and step two of step one are presented in sections 4.5 and 4.4 respectively however such sections don’t exist in the paper. Lines 239 and 245 also refer to subsections of 4, authors probably meant 2.4 and 2.2 respectively.

Line 243, the formula seems to be out of context. Better to introduce the formula before the explanation. Most of the formulas in page 8 are lacking proper context. It could be an alignment issue but only the loss function is connected to the paragraph. Line 286 has an issue with the symbols that represent the word pairs. The formulas on page 9 are also not properly introduced or explained.

The prompt for Figure 3 is not consistent with other prompts that provide a brief explanation of the diagram. The introduction of Figure 3 in the text is also abrupt and out of context (line 246)

The sentence at 304 is unclear in meaning.

Section 3 is lacking a brief introduction to the section.

The tables with results for the experiments show absolute percentage differences. The text would benefit from showing relative percentage improvements. A bar graph might be appropriate for aggregating all this information.

Experimental design

The experiments are well framed and the authors went above and beyond to show the soundness of their results. Also, the code provided in the raw data is complete and clear. The two datasets that were used (MAVEN-ERE and ACE 2005) are popular enough to provide a common framework for comparing with other approaches. Also, reporting on 4 different metrics takes away any concern about accidental improvement. Additionally, the model is tested without filtering and without the graph extractor to show the impact that each of these components has on the overall performance.

Validity of the findings

It is clear that the approach as a whole is an improvement in clustering metrics over other similar approaches. When the different components of the proposed system (like filtering and extraction) are removed, then the framework's performance fails to surpass the closest clustering method (ETypeClus). However, each component of the clustering framework has been carefully designed to fit into the method as a whole so it would be hard to evaluate what is causing the biggest gain in performance.

Additional comments

Overall the paper presents a sound and well-tested approach to event schema induction. The text needs considerable work to make it more accessible and free of errors. Otherwise, the idea is novel enough and with merit and I appreciate the well structured code and raw data included.

Cite this review as

Reviewer 2 ·

Basic reporting

Introduction is not clear:
- Does not seem self-contained, using several terms not previously introduced.
- The definition of the event schema induction problem is not easy to grasp in the text, which does not include enough examples and citations.
- The motivation for the proposed method and its design choices seems based on vague claims, without clear justification or references to back them up.
- The list of main contributions of the study does not seem to mention contributions but rather mere tasks that were carried out.

Typos and formatting errors are present in several parts of the text.
The term "graph" is mentioned in the text several times in a vague and inconsistent way. I would suggest reviewing and making this more clear and formal.

Experimental design

The section that introduces the methodology is very difficult to follow and seems overly complicated. A series of adopted techniques are listed, without clear justification or intuitive explanation. It does not always make it clear which ideas are original or adapted from previous work. All of this makes it difficult to assess the technical merit and correctness of the methodology, as well as its independent reproduction.

I don't think the text clearly defines the research question or how it is approached.

Validity of the findings

The results seem to show the potential of the proposed methodology. However, given the previous comments, it is difficult to verify its validity.

The data appears to have all been made available, but the conclusions are not technically clear given the previous comments again.

Cite this review as

---

## Round 0.2 · Minor Revisions

The second round of reviews are positive in the overall structure and novelty of the work. One of the reviewers has pointed out an array of fixable points that can be addressed in a moderate timeframe. I am recommending a minor revision and ask the authors to address these and resubmit. Please take a look at the list provided by the one reviewer who has highlighted issues and suggested how to correct them.

·

Basic reporting

The paper has been greatly improved since its last revision. Some semantic and grammatical errors persist but can be easily resolved. As follows is a list of issues that were found in the text:

Abstract:

- "Existing event schema induction often..." I believe the text is missing the word methods or models after schema.
- If the method uses only four metrics it might be best to list them
- "... overlap and acceptable metrics", overlap of what exactly and what constitute acceptable metrics?. This is defined in the paper but it is an unclear term in the abstract.
- Last sentence makes redundant use of the word "show" and makes excessive use of passive voice.

Body:

- Line 27, the word all is redundant
- Line 53, if this approach is addressing the limitation of Shen 2021, phrase it as such.
- Line 61, add citation to "Graph Autoencoder".
- Line 74, clustered implies that the extracted clusters are clustered again. This is not clear in the text so most likely clustered is being used redundantly
- Line 76, "the four metrics" is used throughout the document before being defined. I suggest either listing the metrics earlier by name or add a sentence alluding to the section where they are described.
- Lines 83-88, the main contribution is the graph construction method, and the other two sentences support the approach but are not separate contributions. I suggest restructuring this list to show that
- Line 86, well done is ambiguous.
- Line 87, might be best to change it to "Experiments show that..."
- Line 101, "Bootstrapping..." is a fragmented sentence. It is unclear what the sentence is trying to communicate.
- Line 102, the object of the paragraph is people. If most refers to people, the sentence has little meaning. Specify what most refers to.
- Line 173 "In this section.." is not needed.
- Line 173-175 rephrase the sentences using present indicative or preterite verbs instead of infinitive. Step 3 is not a complete sentence
- Line 197, the features are not constructed in the document. Rephrase the sentence. I suggest "In this section we describe how we construct..."
- Line 202, "concatenate" instead of "concat"
- Line 209, how many associations were excluded? This step makes it hard to reproduce the results in the paper, an external reference to the relationships that where considered might benefit the reader.
- Figure 3, there's an issue with the border of the picture which cuts the text in half (reviewing the pdf)
- Line 313 "We following metrics.." unclear what the sentence is trying to convey, might be missing a word or two.
- 314-319 Overlap metrics and Acceptable metrics implies that each one is a set of metrics. However, the description portrays a single metric for each. The name for these metrics seem to come from the authors due to lack of citations, maybe clarify that. However, these seem to be ratios so Overlap Ratio and Acceptable Ratio seem like more appropriate names for them.
-(optional) Line 346-347, improvement is expressed as relative but the numbers are absolute. I suggest adding relative improvement in parenthesis
- Ablation study: Change every mention of deleting a component to removing the component. For example, Removing the graph autoencoder...
- Figure 4 is too small and unnecessary. Discussion of how partial GESI compares to other methods is irrelevant to the Ablation Study, the discussion should be centered on the components themselves.
- Lines 367 -373, these observations are better drawn from tables 4 and 5 than from Figure 4.
- Lines 380-392 should not be part of the list of observations as this is the insight resulting from those observations. Unindent that paragraph.
- Tables 6 and 7 would be best described as a sample of the results instead of partial result.
- Table 8, can the overlap and acceptable percentages be calculated for the other methods?
- Line 399, the use of the word "prove" seems inappropriate since it is evaluated within a method designed for it. "show" or "strongly indicate" would be more appropriate.
- Abstract introduction and conclusion: "highly acceptable metrics" is a confusing statement. I believe the authors are communicating that the model performed well on the "acceptable" metric they devised. They need a different name for these metrics because the names overlap and acceptable create semantic confusion.
- Line 405, direction (1): Provide insight as to why that would yield better results.
- Line 407, direction (3): This is a run-on sentence, use proper separation for the two statements (period or semicolon).

Experimental design

The experiments are sound, thorough, and well represented in tables and figures. Figure 4 is the only one with issues as it is hard to read and presents information that is already easier to read in the preceding tables.

The authors present two metrics: overlap and acceptable. The naming of these metrics is unfortunate for it generate sentences that are confusing to read. The lack of citations or common reference to these metrics point to them being coined by the authors. The description of the metrics is clear and can easily be seen as necessary. However, it would be best to include the other methods in the literature for this test so we can compare.

Using the phrase "highly acceptable" as a way to describe a high percentage in the acceptable metric is a misleading statement. It should be fixed after fixing the name of this metric.

Validity of the findings

The new methods beats the state of the art on well known clustering metrics and the ablation study shows that each component deviced by the authors is contributing to that performance. The use of new metrics (overlap and acceptable) is only acceptable because of this context. The observations provided in the conclusion also show insight and clear paths for improvement. Other than the issues pointed out already I believe this work does show a novel approach that outperforms similar methods.

Cite this review as

Reviewer 2 ·

Basic reporting

I appreciate the authors' thoughtful responses to my comments. I believe the article has been significantly improved. The text appears clearer and most issues have been addressed. I still find the list of contributions at the end of the introduction difficult to understand (removing it doesn't seem like a bad option). The motivation and definition of the problem are also clearer and accompanied by more adequate/complete references.

Experimental design

no comment

Validity of the findings

no comment

Cite this review as

---

## Round 0.3 · accepted · Accept

The authors have addressed the remaining minor issues.